# Knowledge, perceptions and attitude of Egyptian physicians towards biobanking issues

**Ahmed Samir Abdelhafiz**[1]*, **Eman A. Sultan**[2], **Hany H. Ziady**[2], **Douaa M. Sayed**[3], **Walaa A. Khairy**[4]

**1** Department of Clinical Pathology, National Cancer Institute, Cairo University, Cairo, Egypt, **2** Department of Community Medicine, Faculty of Medicine, Alexandria University, Alexandria, Egypt, **3** Department of Clinical Pathology, South Egypt Cancer Institute, Assiut University, Assiut, Egypt, **4** Department of Community Medicine, Faculty of Medicine, Cairo University, Cairo, Egypt

* ahmed.samir@nci.cu.edu.eg

**Data Availability Statement:** All relevant data are within the paper and its Supporting Information files.

## Abstract

### Objectives

Collection and storage of biospecimens and data for biobanking raise many ethical concerns. Stakeholders' opinions about these ethical issues are important since they can help in the development of ethical guidelines to govern biobanking activities. Physicians are among the important stakeholders since they contact potential participants and could be biobank users. The goal of this study is to evaluate the perceptions and attitude of Egyptian physicians towards ethical issues in biobanking.

### Methods

A cross-sectional online survey was designed and distributed with the target group between November 2019 and January 2020.

### Results

The questionnaire was completed by 223 physicians. While 65.5% reported hearing the term "Biobanking" before, 45.7% knew that there are biobanks in Egypt. Participants had a general positive attitude towards the value of biobanks in research. About 73% agreed that biobanks can share biospecimens with international research organizations, but only 42.6% supported collaboration with pharmaceutical companies, and 44% agreed to the use of user fees by biobanks. About 48% supported the use of broad consent in biobanks, and 73.1% believed that donors of biospecimens should be informed about results of research performed on their biospecimens.

### Conclusion

Although many Egyptian physicians heard about biobanking, they had limited knowledge about the existence of biobanks in Egypt. They had concerns about commercialization, use of broad consent and user fees. A knowledge gap exists among these stakeholders, which

**Funding:** The author(s) received no specific funding for this work.

**Competing interests:** The authors have declared that no competing interests exist.

should be covered by different educational activities. Community discussions should start to reach consensus about the issues of commercialization and return of research results.

## Introduction

The complete mapping of the human genome has greatly expanded our ability to better understand the genetic basis of human disease and to improve management, thus ushering in new era of personalized medicine [1, 2]. This development has led to the need for the establishment of biobanks which are able to collect and supply large amounts of samples and data required to conduct large genomic studies [3].

Genomic research supported by population biobanks is growing worldwide, guided by the hope that this research will lead to better disease prevention and treatment through novel drug targeting, personalized treatment, as well as prediction of disease risk [1, 2, 4, 5]. In addition to population-based biobanks, human biobanks can be disease-based (collecting samples from patients with a specific disease or a group of related diseases), genetic (DNA/RNA), project-driven, or tissue versus multiple specimen types. They can be non-profit, commercial, or even virtual (a virtual biobank is an electronic database of biospecimens and data that exists independent of the location of the stored samples) [5–9].

Major changes in the research environment and how research is conducted, including the growth of multi-institutional projects with large amounts of data being shared, as well as the concomitant advances in genomic technologies, have raised new ethical questions [10]. These include, among others, questions about privacy and confidentiality, protection of human subjects, benefit sharing, protection of innovation and intellectual property rights, commercialization, and the ethics of genomic data-sharing [10, 11].

In Egypt, eight disease-based biobanks have been founded in the past few years in order to support both clinical and genomic research [12]. Locations of these biobanks were chosen with aim of them being equally geographically distributed. They are located in six governorates, starting with Alexandria in the north to Luxor and Aswan in southern Egypt. All of them are non-profit and are affiliated to universities or nongovernmental organizations. Although the number of biobanks and their research activities are growing in Egypt, no specific guidelines or regulations for biobanks have been formulated. In Egypt, the only 'guidelines' to biobank related research may lie in a number of chapters within a few medical documents [13]. Ethical, legal, and social issues (ELSI) are currently among the hottest topics in biobanking research. The absence of full ethical and regulatory frameworks that would govern genomic studies and biobanking activities represents one of the most pressing research-related ethical challenges in many African countries, including Egypt [14]. Most research ethics committees (RECs) in Egypt use international research ethics guidelines, such as the Declaration of Helsinki and guidelines of the Council for International Organizations of Medical Sciences (CIOMS) and/or the Islamic Organization for Medical Sciences (IOMS) to review research protocols [15].

While biobanking is growing in Egypt, newly developed biobanks aim at achieving long-term sustainability, which represents one of the biggest challenges for biobanks [16]. Social, operational, as well as financial aspects should be taken into consideration when deliberating plans aimed at achieving the sustainability of a biobank [17, 18]. Social sustainability, which focuses on maintaining the acceptability of biobanks and their activities among their potential stakeholders through proper engagement, is among the important ELSI issues [17].

Biobank stakeholders include participants, physicians, researchers, research institutions, funders, as well as ethics committees [2, 16]. It is important to formulate proper strategies that would aid in identifying, communicating with and engaging all potential stakeholders in order to achieve mutual benefit [18]. The development of local ethical guidelines to govern biobanking should take into consideration the views of different stakeholders. The first step of building a good relationship between biobanks and their different stakeholders is to survey their prior knowledge and attitudes with regards to the concept of biobanking and related ethical issues.

In a previous work, we reported on the attitudes and prior knowledge of Egyptian patients as potential stakeholders of biobanking [7]. Building on our first report that focused on patients, we examined, in this work, the perceptions, attitudes and knowledge of Egyptian physicians of biobanking and biobanking activities in Egypt.

## Subjects and methods

### Study design, participants, sampling and setting

A cross-sectional online survey was designed to for the purposes of this study. The target population was Egyptian physicians currently working in Egypt at different public and private health centers, who work in three governorates representing the three main geographical regions of Egypt; the north (Alexandria governorate), the capital or "the centre" (Cairo governorate), and the south (Assiut governorate). Sample size was determined using the Epi Info 7 software based on the expected probability of positive attitude of health professionals towards biobanking (82.9%) [19] at a 95% confidence interval, limit of precision of 5%, with a design effect of 1.0. The calculated sample size stood at 218 participants.

### Study tool and data collection

An online English-language questionnaire was constructed. An introduction preceded the first section of the questionnaire on Google Forms. The introduction included a brief explanation of what biobanks were and a summary of their role in research. This was supplemented by a number of links to videos and sources of further reading about biobanks (S1 File). The survey questionnaire was comprised of three main sections; a) sociodemographic information; b) basic prior knowledge of biobanking; and c) perceptions and attitudes towards biobanking. Participants were asked to record their level of agreement with survey items (statements) by using a 5-point Likert scale (strongly agree = 5, agree = 4, unsure = 3, disagree = 2 and strongly disagree = 1).

A preliminary phase was conducted in order to assess the validity and reliability of the questionnaire before its wider use. For the assessment of validity, three Egyptian experts in the field of biomedical research were asked to assess the degree to which items in the questionnaires were relevant and could correctly measure prior knowledge and attitudes of physicians towards biobanking. After that, corrections were done as per the experts' remarks. For the assessment of reliability, 15 physicians were asked to respond to the questionnaire twice. Responses were collected three weeks apart. The results showed an adequacy of both internal consistency reliability (Cronbach's alpha = 0.78), and test-retest reliability (intra-class correlation coefficient = 0.97) [20].

During the data collection phase, participants were invited to participate in the study via e-mails and through different social media platforms. The participants were recruited using the convenient sampling technique, which facilitated reaching the required sample size within the available time and resources. The questionnaire was designed on Google Forms, and those who agreed to participate were asked to use the link in the invitation e-mail/message to reach the questionnaire. The purpose of the survey was explained both in e-mail messages and posts

made on social media pages. A four-month period was required to achieve the required sample size. Data were collected between November 2019 and January 2020.

## Statistical analysis

All data analyses were performed using the Statistical Package for the Social Sciences (SPSS) software, version 22. Data were presented as numbers and percentages for categorical variables, and as means and standard deviations (SD) for quantitative continuous variables. The chi-square test was used for the purposes of testing associations between qualitative variables. In case of expected counts below 5, Fisher's exact test was used. All results were interpreted at a 5% level of significance.

## Ethical considerations

The study was performed in accordance with relevant regulations [21]. Invited physicians were asked to carefully read the provided text in the Google form to fully orient themselves with the purpose of the survey and the relevant background information regarding biobanks. Informed consent was obtained using the following statement, which was shown to potential participants before starting to answer the questionnaire: "Completing this questionnaire and submitting it will be considered as your informed consent to participate in this study. You have the right to stop completing the questionnaire at any stage and to withdraw from participation." Those who responded to the questionnaire to completion were considered as having given their informed consents. Official approvals for implementation of the study were obtained from the Ethical Committee of the Faculty of Medicine at Alexandria University. Confidentiality of data was ensured throughout all phases of the study. Data were analyzed anonymously, with only members of the study in charge of data analysis having access to collected data. Confidentiality of data continued until the full manuscript was finalized. After publication, the data will be safely stored with continued maintenance of confidentiality.

## Results

The questionnaire was completed and submitted by 223 out of 291 physicians, yielding a response rate of 76.6%. About half of the participants (47.1%) were from Cairo, followed by Alexandria (37.7%) and Assiut (15.2%). Demographic and biobanking related characteristics of the respondents are summarized in S1 Table. Nearly half the respondents (43%) reported that blood sampling was part/would be part of their current or future research, while only 38.1% and 25.1% of participants gave analogous responses with regards to tissue and saliva/urine samples respectively. Most physicians (79.4%) reported having never attended any type of scientific event that had discussed biobanking.

Physicians' responses to the 'basic knowledge' items are detailed in Table 1. About two thirds of physicians reported having heard the term "biobanking" before. However, only 45.7% had knowledge about the existence of a number of biobanks in Egypt.

**Table 1. Basic knowledge about Biobanking among the respondents (n = 223).**

| Knowledge item | Number | Percent |
|---|---|---|
| I have heard the term "Biobanking" before | 146 | 65.5 |
| There is a biobank in my institute | 69 | 30.9 |
| There are several biobanks in Egypt | 102 | 45.7 |
| There is a law that governs biobank work in Egypt | 114 | 51.1 |

**Table 2. Level of agreement of the respondents to the survey items regarding general attitudes towards biobanking (n = 223).**

| Survey Items | Level of Agreement | | | | | | | | | |
|---|---|---|---|---|---|---|---|---|---|---|
| | Strongly Agree | | Agree | | Unsure | | Disagree | | Strongly Disagree | |
| | N | % | N | % | N | % | N | % | N | % |
| 1. I think that the presence of biobanks is important for the development of new treatments | 115 | 51.6 | 98 | 43.9 | 9 | 4.0 | 0 | 0.0 | 1 | 0.4 |
| 2. I think that the presence of biobanks is important for the development of new methods of diagnosis | 114 | 51.1 | 96 | 43.0 | 12 | 5.4 | 0 | 0.0 | 1 | 0.4 |
| 3. I think biobanks can make a difference in biomedical research in general | 116 | 52.0 | 98 | 43.9 | 8 | 3.6 | 0 | 0.0 | 1 | 0.4 |
| 4. I think biobanks can make a difference in aspects related to quality of samples and data provided for research | 104 | 46.6 | 102 | 45.7 | 16 | 7.2 | 0 | 0.0 | 1 | 0.4 |
| 5. In the future, I will be interested in applying to get samples for my research from the biobank | 78 | 35.0 | 96 | 43.0 | 46 | 20.6 | 2 | 0.9 | 1 | 0.4 |
| 6. I would be willing to help create a biobank in my institute | 81 | 36.3 | 91 | 40.8 | 47 | 21.1 | 1 | 0.4 | 3 | 1.3 |
| 7. I will donate samples myself and will ask my relatives to donate samples to the biobank | 45 | 20.2 | 79 | 35.4 | 87 | 39.0 | 8 | 3.6 | 4 | 1.8 |
| 8. If there is a biobank in my institution, I will inform my patients about it | 80 | 35.9 | 108 | 48.4 | 31 | 13.9 | 2 | 0.9 | 2 | 0.9 |
| 9. I would like more information about biobanks to be more readily available | 141 | 63.2 | 76 | 34.1 | 5 | 2.2 | 0 | 0.0 | 1 | 0.4 |
| 10. If there is a course/ lecture/conference about Biobanking, I will be interested in attending | 77 | 34.5 | 113 | 50.7 | 31 | 13.9 | 0 | 0.0 | 2 | 0.9 |
| 11. Donating samples for research is in line with religious beliefs | 61 | 27.4 | 88 | 39.5 | 67 | 30.0 | 4 | 1.8 | 3 | 1.3 |

Table 2 summarizes participants' general attitudes towards biobanking. The majority of participants' responses towards the survey items indicated positive attitudes towards biobanking (strongly agree–agree). It is noteworthy that physicians reported the highest level of agreement (strongly agree–agree) in response to the following statements "*I would like more information about biobanks to be more readily available*" (97.3%); "*I think biobanks can make a difference in biomedical research in general*" (95.9%); "*The presence of biobanks is important for the development of new treatments*" (95.5%); and "*The presence of biobanks is important for the development of new methods of diagnosis*" (94.1%). Moreover, the following statements yielded the highest level of uncertainty (a response of 'unsure') among participants: *"I will donate samples myself and will ask my relatives to donate samples to a biobank"* (39%); "*Donating samples for research is in line with my religious beliefs*" (30%); "*I would be willing to help create a biobank at my institute*" (21.1%); and "*In the future, I will be interested in applying to acquire samples for research from a biobank*" (20.6%).

Table 3 outlines participants' responses with regards to issues of privacy, data sharing and access to data. The highest level of disagreement, at 41.3% (strongly disagree–disagree), was in

**Table 3. Level of agreement of the respondents to the survey items regarding privacy, sharing and access issues (n = 223).**

| Survey Items | Level of Agreement | | | | | | | | | |
|---|---|---|---|---|---|---|---|---|---|---|
| | Strongly Agree | | Agree | | Unsure | | Disagree | | Strongly Disagree | |
| | N | % | N | % | N | % | N | % | N | % |
| 1. Biobanks can share samples and data with international research organizations | 62 | 27.8 | 101 | 45.3 | 39 | 17.5 | 13 | 5.8 | 8 | 3.6 |
| 2. Biobanks can share samples and data with commercial and pharmaceutical companies | 24 | 10.8 | 71 | 31.8 | 60 | 26.9 | 38 | 17.0 | 30 | 13.5 |
| 3. The biobank may provide any medical information to insurance companies | 16 | 7.2 | 55 | 24.7 | 67 | 30.0 | 50 | 22.4 | 35 | 15.7 |
| 4. The biobank may provide any medical information to treating physician | 56 | 25.1 | 127 | 57.0 | 29 | 13.0 | 6 | 2.7 | 5 | 2.2 |
| 5. The biobank can provide any medical information to government | 23 | 10.3 | 77 | 34.5 | 62 | 27.8 | 34 | 15.2 | 27 | 12.1 |
| 6. The biobank can provide confidential medical information to legal authorities if asked | 28 | 12.6 | 97 | 43.5 | 48 | 21.5 | 28 | 12.6 | 22 | 9.9 |
| 7. I will not donate my samples to a biobank because my identity could be known through my DNA | 10 | 4.5 | 35 | 15.7 | 86 | 38.6 | 74 | 33.2 | 18 | 8.1 |

**Table 4. Level of agreement of the respondents to the survey items regarding governance (n = 223).**

| Survey Items | Level of Agreement | | | | | | | | | |
|---|---|---|---|---|---|---|---|---|---|---|
| | Strongly Agree | | Agree | | Unsure | | Disagree | | Strongly Disagree | |
| | N | % | N | % | N | % | N | % | N | % |
| 1. A participant who donates blood for scientific research on genes and environment remains in control over his/her blood | 22 | 9.9 | 81 | 36.3 | 72 | 32.3 | 42 | 18.8 | 6 | 2.7 |
| 2. Biobanks owns the stored samples | 14 | 6.3 | 65 | 29.1 | 71 | 31.8 | 59 | 26.5 | 14 | 6.3 |
| 3. Biobanks are just in custody of the samples, but don't own them. | 34 | 15.2 | 108 | 48.4 | 50 | 22.4 | 28 | 12.6 | 3 | 1.3 |
| 4. Biobanks may charge user fees for samples that are distributed to researchers. | 12 | 5.4 | 86 | 38.6 | 63 | 28.3 | 45 | 20.2 | 17 | 7.6 |
| 5. A transparent policy for distribution of samples to researchers should exist. | 124 | 55.6 | 86 | 38.6 | 10 | 4.5 | 1 | 0.4 | 2 | 0.9 |

response to the following statement "*I will not donate my samples to a biobank because my identity could be known through my DNA*". On the other hand, the following statements yielded the highest levels of agreement (strongly agree–agree): "*Biobanks may provide any medical information to the treating physician*" (82.1%); and "*Biobanks may share samples and data with international research organizations*" (73.1%).

Responses to statements regarding issues of governance are presented in Table 4. Out of five survey statements, the majority of the physicians (94.2%) supported (strongly agree–agree) the following statement: "*A transparent policy for distribution of samples to researchers should exist*". On the other hand, the statement asserting that "*biobanks own the stored samples*" attained the lowest level of agreement (35.4%) of the five statements in the governance section of the questionnaire.

With regards to consent and biobank participant's rights, most responses revealed a general agreement among physicians, as shown in Table 5. The following survey items yielded the highest level of agreement (strongly agree–agree) "*Participant's information and samples stored in the biobank should be protected and securely stored*" (96%); "*Sample donors should be informed in details of how their samples will be used*" (84.8%); "*Sample donors should be informed if their samples will be transferred abroad*" (78.1%). *Sample donors should be compensated by some means for their samples*" (48.5%); and "*A broad consent that does include every future research is more suitable for biobank work*" (47.9%).

Associations between demographic data and biobanking related variables and knowledge of biobanking are summarized in S2 and S3 Tables. Prior knowledge of the term "biobank" was significantly associated with two variables; gender (P = 0.019) and having attended a scientific event that discussed biobanking (P<0.001). No significant association was found between

**Table 5. Level of agreement of the respondents to the survey items regarding consent and participant's rights (n = 223).**

| Survey Items | Level of Agreement | | | | | | | | | |
|---|---|---|---|---|---|---|---|---|---|---|
| | Strongly Agree | | Agree | | Unsure | | Disagree | | Strongly Disagree | |
| | N | % | N | % | N | % | N | % | N | % |
| 1. A broad consent that does not include every future research is more suitable for biobank work. | 34 | 15.2 | 73 | 32.7 | 82 | 36.8 | 24 | 10.8 | 10 | 4.5 |
| 2. Participant information and samples stored in the biobank should be protected and securely stored. | 159 | 71.3 | 55 | 24.7 | 6 | 2.7 | 1 | 0.4 | 2 | 0.9 |
| 3. Sample donors should be compensated by some means for their samples | 28 | 12.6 | 80 | 35.9 | 68 | 30.5 | 40 | 17.9 | 7 | 3.1 |
| 4. Sample donors should be informed in detail how their samples will be used | 105 | 47.1 | 84 | 37.7 | 19 | 8.5 | 11 | 4.9 | 4 | 1.8 |
| 5. Sample donors should be informed if their samples will be transferred abroad | 111 | 49.8 | 63 | 28.3 | 32 | 14.3 | 14 | 6.3 | 3 | 1.3 |
| 6. Sample donors should be informed about results resulting from research on their samples | 82 | 36.8 | 81 | 36.3 | 44 | 19.7 | 16 | 7.2 | 0 | 0.0 |

knowledge of the term "biobank" and age, residence, years of experience, affiliation, or specialty. There was no significant association between knowledge of biobanking and current or future research interests of the participant.

## Discussion

The number and scope of activities of human biobanks have rapidly expanded in the last 20 years. Current genomic research involves the use of large numbers of samples collected from many participants in order to reach statistically significant healthcare-related conclusions [6]. In order to properly play their needed role in research, biobanks should be run under an effective governance framework. This framework should include rules and practices by which biobanks would ensure accountability, fairness, and transparency to funders, patients, researchers, as well as the society at large. Development of stakeholder engagement strategies is an important component of biobank governance [22]. Physicians are among the most important biobank stakeholders due to several reasons. Firstly, they communicate with patients, earn their trust, can directly educate them as to the importance of research and can ask for their permission to be contacted by biobank personnel [23]. Secondly, most physicians in Egypt have to engage in some level of research in the form of masters and/or doctorate research in order to attain specialty and sub-specialty certification. Such research may necessitate the use of biobanking services. Finally, all biobanks in Egypt are currently disease-based biobanks. Thus, it is important to engage with physicians and evaluate their perceptions and attitudes towards biobanking related issues, which was the goal of this study.

### 1. Knowledge of biobanking

Overall, responses showed that participants had an acceptable degree of knowledge of the meaning and function of biobanks, with 65.5% of participants indicating that they had heard of the term biobanks before (Table 1). In an earlier comparable survey of patients as stakeholders, we reported that more than 80% of participants had never heard of the term biobanking prior to participating in the survey [7]. This disparity in knowledge between patients and physicians is comprehensible since physicians conduct research and thus become familiar with related terms. A previous study showed that there was a comparatively lower level of knowledge (36%) of biobanking among health professionals in Australia [24]. This difference may be attributed to the differences between study populations since most health professionals in the Australian study were nurses. Despite recognizing the 'term 'biobanking,' less than half of our participants (45.7%) indicated knowledge of the existence of several biobanks in Egypt. This may reflect the lack of a comprehensive marketing strategy at these biobanks. This was also evident from the limited number of participates who reported attending any form of educational activity that discussed biobanking (S1 Table). Knowledge about the term biobanking was significantly higher among those who had attended biobanking-related scientific events (S3 Table). In our previous work, we were able to successfully communicate the concept of biobanking to undergraduate students of life sciences using different tools, including Facebook, the most popular social media platform in Egypt [12]. We recommend using social media platforms for the purposes of marketing the concept of biobanking among physicians as well, since the use of social media has also proven effective in communicating such concepts and related health issues with other stakeholders in Egypt [12, 25].

### 2. General attitude towards biobanking

Results of this study indicated a general positive attitude towards biobanking and recognition of their value in biomedical research (Table 2); a result comparable to that reported by the

study conducted among health professionals in Australia [24]. Despite their positive attitude, only 55.6% of participants in this study were willing to donate their own samples or ask their family members to donate samples to biobanks. Interestingly, a similar proportion (57.7%) of 315 senior Egyptian medical students studying in medical schools in the same governorates (Cairo, Alexandria, and Assiut), who participated in a similar survey indicated a similar reluctance to donate samples or encourage family members to do so (data not published). Another comparable study to the current work reported a negative correlation between willingness to donate samples to biobanks and higher levels of education [26]. Evidently, some physicians as well as medical students prefer to lead the process of patient care and research, but not to take part in it by donating their own samples. This may be due to possible concerns or fears with regards to participating, as subjects, in genetic research. These potential concerns should be explored in future studies, and they should be approached with respect.

Islam is the religion of the majority of Egyptians, which is followed by Christianity. The second article of the Egyptian constitution states that "Islam is the Religion of the State. Arabic is its official language, and the principle source of legislation is Islamic Jurisprudence" [27]. In our study, about 30% of participants were not certain if the teachings of their religion allowed sample donation (Table 2). This was in contrast to the response of the majority of patients in our previous survey in which 86.5% of patients (Muslims and Christians) indicated that donating samples did not go against their religious beliefs. This might reflect the physicians, by nature, usually look for evidence-based opinions in both their practices as well as in their personal lives. Muslim religious scholars have indicated that Islam permits the establishment of research biobanks if autonomy is respected [28, 29]. In general, principles of confidentiality have received significant attention in the Holy Qur'an, the Sunnah (habits, practices, sayings, and commands of the Prophet Muhammad, and in Islamic juristic writings [28]. The website of the Qatar biobank, another majority Muslim country, states that "Islam is permissive when it comes to the use of biological information for research purposes if it is regulated within the context of culture and religion"[30]. All in all, religious beliefs do not appear to represent a barrier to the establishment and work of biobanks in Egypt if the general values of research ethics are respected.

## 3. Perception and attitudes regarding privacy, access, and sharing of samples

Biobanks collect and store samples and data in order to provide them to researchers for use in different research projects. Strong governance and transparent policies regulating access to these sample and data collections should be formulated [31]. In general, access to samples and data should be governed with emphasis on scientific quality of research proposals, prioritizing research of value and ethical soundness [31]. Maintaining proper governance of access is often met by challenges stemming from the different and sometimes conflicting interests of stakeholders, including biobank managers (who may seek academic or financial recognition of their efforts when material or data are used in research); researchers, who pursue their own research interests when they ask for access for samples and data; and funders, who may oblige biobanks to only permit the use of samples and data in research of high scientific and social value [31].

Although sharing different types of samples and data across borders is now a routine part of international collaborative research, several challenges are often associated with this practice. Those include the need to protect the confidentiality of participants, assure benefit sharing, protect intellectual property rights, and decide on who claims authorship of resultant publications [32, 33]. Concerns over sharing samples with international collaborators are

particularly strong in Egypt. A recent law passed by the Egyptian Parliament attempted to set restrictions over the transport of samples abroad [34]. The law was rejected by the president, who objected to some of its articles [35]. The initial passage of this law reflected the fears of policy makers in Egypt of the potential problems associated with sharing samples internationally. Similarly, many members of RECs in Africa reported strong concerns about the practice of exportation and storage of tissue samples at international institutions, citing fears of the possible exploitation of both African researchers and populations [14].

Most participants in our survey, and in contrast to the patients (as stakeholders) in our previous survey [7], supported sample and data sharing with international research organizations (Table 3). However, collaboration with commercial and pharmaceutical companies represented a concern for them, with only 42.6% agreeing to the idea of sharing samples with commercial and pharmaceutical companies. Although commercialization represents one means of achieving financial sustainability [18], it raises ethical questions regarding fair sharing of benefits and represents a concern for biobank sample donors [36]. Concerns about biobank commercialization are linked to fears of losing control of public biobank resources to for-profit companies and organizations, which may stand in contradiction to the altruistic purposes for which participants volunteer to donate [36, 37]. Benefit sharing can provide a solution to this dilemma. If genetic data are used to generate healthcare-related benefits, fair distribution of these benefits should be carried out, and these benefits should reach the community which had supplied the genetic material used to generate data resulting in these benefits [38].

In our previous study, less than 28% of our survey participants supported sharing their potential samples with pharmaceutical companies [7]. We believe that several stakeholders in Egypt are not yet ready to explore commercialization due to fears of possible misuse of samples and data. That is why we recommend that the commercialization of biobanks in Egypt be delayed until these concerns are fully addressed. We also think that sample and data sharing across borders should be regulated by law, which should aim to balance the benefits of sharing samples with potential risks.

Protecting the privacy of biobank participants is one of the jobs of biobanks and was stipulated for in the standard operating procedures (SOPs) of Egyptian biobanks. Accordingly, Egyptian biobanks are held accountable for limiting disclosure and safeguarding the integrity of the information stored in them through the use of coding and anonymization [7]. At the legal level, a recent data protection law formulated to govern the processing and handling of personal data was proposed by the Egyptian government and approved by Parliament. The Law applies to "personal data" as well as data that can be combined with other data to identify an individual. The law is meant to promote the security of personal data which is being processed and stored online. It also sets a legal framework for the regulation of transmission of data to other countries [39]. Clearly, a balance should also be maintained between protecting privacy and confidentiality on the one hand and maximizing benefits to the individual and/or the society at large on the other. Our participants provided a possible framework for such a balance, with most agreeing that a biobank should be able to give access to medical information to treating physicians when needed, but not to the government or to insurance companies. Still, slightly more than 50% of participants believed that biobanks should provide confidential medical information if presented with a proper court order (Table 3). In our study of patients as stakeholders, 71.8% of participants agreed on a similar question regarding legal authorities [7]. Article no 30 of the list of professional ethics of the Egyptian Medical Syndicate states that "A doctor is not permitted to divulge the secrets of his patient which he/she had become acquainted with by virtue of his/her profession, unless such divulgence was decreed by a court order or in the event of possible serious and certain harm to others, or in other cases determined by law" [40]. However, this applies mainly to medical practice and not

research. Thus, conditions where breach of confidentiality during clinical research would be justifiable must be clearly detailed by the syndicate or similar body.

The majority of participants in a survey conducted with different biobank stakeholders in Saudi Arabia, including researchers, physicians, medical students, donors, and members of the general public, agreed that confidentiality might be breached in specific cases with clear justifications. A number of situations they agreed would justifiably warrant such a breach included the presence of infectious and/or genetic disease that may affect others, or the presence of a court order [41]. In a study that aimed at exploring stakeholder perspectives on the use of biological samples for future unspecified research in Malawi, participants emphasized the importance of privacy and confidentiality and believed that it should be a top priority. They also indicated that there was a need for the formulation of guidelines to govern sharing and access to samples [42]. We think that there should be clear justifications for any breach of privacy if samples were to be used for purposes other than research, and this should be regulated by law.

## 4. Perceptions and attitude issues related to governance of biobanks

Millions of human samples are collected and stored in biobanks worldwide. The long-term storage of these samples and associated data raises questions about control and ownership of samples [43]. Most participants in our survey indicated a belief that biobanks are just custodians of the samples, but that they do not own them (Table 4). A study conducted to explore the views of university researchers' in the United States with regards to the ownership of human genetic specimens reported divergent views with regards to the ownership of retained specimens, with many study participants identifying more than one potential owner [43].

A previous study proposed that custodianship would best describe the role of researchers and biobanks with regards to samples [44]. Custodianship is defined as the responsibility of caretaking of biospecimens from initial collection until distribution of research findings whilst following a set of relevant operating principles [44]. This custodial model ensures transparency in research, protection of the rights of research participants, and accountability among all stakeholders involved in biospecimen research [44]. Another study proposed a stewardship model, which stipulates that all members of the research team are responsible for the protection of participants' interests and well-being to the best of their abilities [45]. Far superior to the concept of ownership, we believe that the concepts of custodianship and stewardship best describe the role of the biobanks in relation to the samples they maintain.

To support the growing need for high quality biospecimens and data, biobanks should have a business plan to achieve operational, social, and financial sustainability. Financial sustainability may be achieved through institutional support, together with the acquisition of short-term and long-term grants [18]. However, these methods of financial sustainability are not always readily available. Biobanks should thus consider cost recovery models and strategies, including biospecimen user fees for access to human specimens and data, and sales of services provided by these biobanks [46]. The concept of assigning user fees is different from commercialization, that latter of which has been a cause of significant controversy [47]. Nevertheless, less than 50% of medical doctors participating in our survey agreed to the concept of user fees. We think that the majority of them confused user fees with the selling/commercialization of samples, which reflects a limited understanding of the differences between the two models.

## 5. Perceptions regarding consent and participants' rights

Biobanks store biospecimens and data for future unspecified research projects. Appropriate consent for sample donors must explain the scope and breadth of consent together with different elements of the biobank's governance framework [48]. Several types of consent could be

used by biobanks. These include, among others, specific consent (specific for a single research project), dynamic consent (using technology to allow participants to choose between broad consent or to approve a single study at a time), and tiered consent (where the participant allows some uses of the samples only and renewal of consent is needed for other studies [10, 47, 49]. A broad consent form that contains adequate details is regarded as a suitable model of consent for use in biobanks [50]. Broad consent should detail the nature of the biobank and types of samples and data to be collected [32, 51–53]. It should be noted that broad consent is not equivalent to blanket consent, in which no details would be offered with regards to possible uses of samples. Lack of clarity about the differences between broad consent and blanket consent was a source of worry among RECs members in African countries [14].

In our study, we chose to only ask participants about broad consent. This was done for two important reasons. Firstly, we believe that broad consent is the most suitable model of consent for Egyptian biobanks, being low-cost and flexible. Secondly, board consent is the only model of consent that has been adopted and approved by different biobanks in Egypt. In our survey, less than fifty percent of our participants agreed to the use of broad consent in biobanks (Table 5). We think that the majority of our participants had no prior knowledge of this type of consent and assumed that it would infringe on the right of sample donors to know how their samples would be used. Again, this reflects the knowledge gap which needs to be filled by biobankers in Egypt.

Another potential right of donors is the right to know results of research conducted using their samples [54]. The majority of participants in this survey, like patients in our previous study [7], believed in this right. A survey conducted to assess the opinions of stakeholders regarding the return of individual research results among researchers, physicians, and the general public in Saudi Arabia showed that there was a strong agreement among participants that sample donors have the right to learn the results of research conducted on samples they had donated if they wished, whether they would benefit directly or indirectly from knowing these results [55].

Managing the return of research results and incidental findings is a complicated issue that raises a lot of ethical questions. Examples of these questions are questions of what should be returned and what results should be considered actionable versus those that should be labeled non-beneficial [55]. Another question is related to how results, which may include complex genetic findings, would be explained to a segment of donors. Such findings could be associated with stigma against them, or affect their ability to secure medical insurance, employment, or proper healthcare [56, 57]. The majority of Saudi health care professionals participating in a study that evaluated these issues agreed on the importance of returning research results and highlighted a number of possible challenges. These included a donor's possible emotion/psychological state; possible difficulties in achieving proper communication between donors and biobank representatives; questions about communicating uncertainties with regards to some research results; and questions with respect to funding of the costs of returning results [54]. We believe that, for the time being, return of results should only be offered in the form of publications and not on individual bases. This should be clearly stated in consent forms. We also recommend the start of a broad discussion among concerned parties in Egypt of the best practices with regards to return or results. It is also essential to conduct training programs for genetic councilors on how to effectively communicate results with participants/patients.

## Conclusions and recommendations

Although Egyptian physicians displayed an acceptable level of knowledge of biobanking, they had limited knowledge of the actual existence of biobanks in Egypt. They had concerns about

the commercialization of biobanks, the use of broad consent and about user fees. A knowledge gap exists among these important biobank stakeholders, which should be covered by active marketing and continuous engagement by biobankers in Egypt. Broad community-wide discussions are essential in order for stakeholders to reach consensus with regards to issues of commercialization and return of research results.

Based on the views of stakeholders and taking into consideration specific factors related to the Egyptian cultural norms and legal framework, we believe that biobanking ethical guidelines would be best developed through the collaboration of different RECs. Such collaboration could be facilitated by the Egyptian Network of Research Ethics Committees, which was founded in 2008 in order to help harmonize the work of different Egyptian RECs [58]. The network includes 23 ethics committees, which can work together to develop an ethical framework for biobanking activities in Egypt. The recently established network of Egyptian biobanks also has an important role to play in this aspect.

## Limitations of the study

There have been some limitations to this study. Firstly, we collected data from physicians working in three governorates using a convenient sampling method for reasons of time and resource limitations. Although these governorates represent the main geographical regions of Egypt, and medical curricula in different medical schools are comparable, differences in research environments at these medical schools could have affected responses. Secondly, no attempt was conducted to ascertain the amount of bioethics education participants had received before recruiting them into the study. Bioethics education could have affected the responses of some participants, especially with regards to the different types of informed consent. Finally, conducting the survey online may have caused a degree of bias in favor of younger physicians, who are more likely to be comfortable using the internet and social media. Due to these limitations, this study would be regarded as an exploratory study rather that a nationally representative survey.

## Supporting information

**S1 Table. Background characteristics of the respondents (n = 223).**
(DOCX)

**S2 Table. The relation between knowledge about biobanking and demographic characteristics of the respondents.**
(DOCX)

**S3 Table. The relation between knowledge about biobanking and biobanking related variables of the respondents.**
(DOCX)

**S1 File.**
(DOCX)

## Author Contributions

**Conceptualization:** Ahmed Samir Abdelhafiz, Walaa A. Khairy.

**Data curation:** Ahmed Samir Abdelhafiz, Eman A. Sultan, Douaa M. Sayed, Walaa A. Khairy.

**Methodology:** Eman A. Sultan, Hany H. Ziady.

**Supervision:** Ahmed Samir Abdelhafiz.

**Writing – original draft:** Ahmed Samir Abdelhafiz, Eman A. Sultan, Walaa A. Khairy.

**Writing – review & editing:** Ahmed Samir Abdelhafiz.

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
