## [Decision Letter · Decision Letter 0]

15 Dec 2020

PONE-D-20-34081

Knowledge, Perceptions and Attitude of Egyptian Physicians towards Biobanking Issues

PLOS ONE

Dear Dr. Abdelhafiz,

Thank you for submitting your manuscript to PLOS ONE. After careful consideration, we feel that it has merit but does not fully meet PLOS ONE’s publication criteria as it currently stands. Therefore, we invite you to submit a revised version of the manuscript that addresses all the points raised during the review process.

A rebuttal letter that responds to each point raised by the academic editor and reviewer(s). Please include page and line number where the revised text is shown, and copy the revised text also to your response. You should upload this letter as a separate file labeled 'Response to Reviewers'.A marked-up copy of your manuscript that highlights changes made to the original version. You should upload this as a separate file labeled 'Revised Manuscript with Track Changes'.An unmarked version of your revised paper without tracked changes. You should upload this as a separate file labeled 'Manuscript'.

We look forward to receiving your revised manuscript.

Kind regards,

Helena Kuivaniemi, MD, PhD

Academic Editor

PLOS ONE

Journal Requirements:

Additional Editor Comments (if provided):

Please clarify how the participants were selected? Does Egypt have only 291 physicians in total? Do the participants represent physicians serving both rural and urban communities?

Some of the cells in the tables have a really small number of responses. Was the statistical analysis different for samples <5 answers?

In the discussion, unpublished results on medical students are mentioned and compared to the current study. Please provide some more details e.g. sample size, which medical schools, how many years of medical school, sex etc. to make it possible to assess if the comparison is relevant.

I suggest including Tindani et al. BMC Medical Ethics 2019 as a reference.

Formatting problems:

1) References: some journal names have not been abbreviated properly.

2) Suppl. Tables 2 and 3 run over the page on the right side. Please use landscape page format for wide tables.

Reviewers' comments:

Reviewer's Responses to Questions

**Comments to the Author**

1. Is the manuscript technically sound, and do the data support the conclusions?

Reviewer #1: Yes

Reviewer #2: Yes

2. Has the statistical analysis been performed appropriately and rigorously? 

Reviewer #1: Yes

Reviewer #2: Yes

3. Have the authors made all data underlying the findings in their manuscript fully available?

Reviewer #1: Yes

Reviewer #2: Yes

4. Is the manuscript presented in an intelligible fashion and written in standard English?

Reviewer #1: Yes

Reviewer #2: No

5. Review Comments to the Author

Reviewer #1: See attachment. The manuscript was a joy to read. It was concise, to the point and of value. The formulation of the study and design was well conceptualized and the paper itself was very well written and logical. There is some minor suggestions within the comments. Please see attachment for specific comments at specific sections.

Reviewer #2: This is a very useful manuscript as LMICs, in which biobanks and genomics research facilitated by biobanks are growing. The paper presents very useful findings from a stakeholder group and makes insightful recommendations on ways to move the discourse forward towards policy. There are some suggestions provided however largely to improve clarity of the arguments presented. In addition, there some instances of use of non-formal word-forms, like don't, that's etc; and perhaps a few grammatical errors. For these, I would suggest that the authors spend a little more time looking at brevity and grammar issues.

6. PLOS authors have the option to publish the peer review history of their article (what does this mean?). If published, this will include your full peer review and any attached files.

Reviewer #1: **Yes: **Carmen Swanepoel

Reviewer #2: **Yes: **Aminu Yakubu

---

## [Author Response · Author response to Decision Letter 0]

25 Jan 2021

Dear Dr Kuivaniemi,

Thank you for your valuable review for our manuscript. Your review really helped us to improve it. As per request of one of the reviewers, the manuscript has been sent for language proofreading. The revised version is attached and changes to the original version are highlighted in yellow. Kindly find our responses to the editor's and reviewers' comments below.

Editor

•Please include additional information regarding the survey or questionnaire used in the study and ensure that you have provided sufficient details that others could replicate the analyses. For instance, if you developed a questionnaire as part of this study and it is not under a copyright more restrictive than CC-BY, please include a copy, in both the original language and English, as Supporting Information. 

o A copy of the survey in English is attached as a part of supplementary files. The survey was communicated in English only.

•Please provide additional details regarding participant consent. In the ethics statement in the Methods and online submission information, please ensure that you have specified what type you obtained (for instance, written or verbal, and if verbal, how it was documented and witnessed). If your study included minors, state whether you obtained consent from parents or guardians. If the need for consent was waived by the ethics committee, please include this information.

o Since the survey was distributed through the internet, informed consent process was described as follows " Informed consent was obtained using the following statement, which was shown to potential participants before starting to answer the questionnaire: “Completing this questionnaire and submitting it will be considered as your informed consent to participate in this study. You have the right to stop completing the questionnaire at any stage and to withdraw from participation.” Those who responded to the questionnaire to completion were considered as having given their informed consents."

Additional Editor Comments (if provided):

•Please clarify how the participants were selected? Does Egypt have only 291 physicians in total? Do the participants represent physicians serving both rural and urban communities? 

o As mentioned in the methods section (Study design, Participants, Sampling and Setting & Study tool and data collection), the study participants were selected using the convenient sampling technique. This technique was used as it is easy to reach the required sample size within the available time and resources. The investigators made communications with focal point physicians who work in the three selected governorates (Alexandria, Cairo and Assiut). These three governorates geographically represent three geographic regions in Egypt; (north, centre and south regions). The investigators sent invitation links by email to the focal points. Similarly, these, by their turn invited more physicians from their governorates. Moreover, Facebook and other social media platforms were also used to recruit participants, provided that they work in the three selected governorates. The selected sample of physician was determined according to the calculated sample size using Epi-Info software. Our sample has no relation with the total number of physicians in Egypt which exceeds 200,000 as per official statistics. It is expected that the participant physicians serving urban communities are more represented in our sample, as all our focal points in the three governorates were serving urban communities.

•Some of the cells in the tables have a really small number of responses. Was the statistical analysis different for samples <5 answers?

o In this case, the association between qualitative variables was tested by using the Fisher's Exact Test instead of chi square test. A sentence was added to the revised manuscript to clarify this point (page 8: Subjects and methods/ Statistical analysis section/ line 5)

•In the discussion, unpublished results on medical students are mentioned and compared to the current study. Please provide some more details e.g. sample size, which medical schools, how many years of medical school, sex etc. to make it possible to assess if the comparison is relevant. 

o The required data has been added as follows " Interestingly, a similar proportion (57.7%) of 315 senior Egyptian medical students studying in medical schools in the same governorates (Cairo, Alexandria, and Assiut), who participated in a similar survey indicated a similar reluctance to donate samples or encourage family members to do so (data not published)".

•I suggest including Tindani et al. BMC Medical Ethics 2019 as a reference.

o The suggested article has been cited three times in the following paragraphs page 5 :"The absence of full ethical and regulatory frameworks that would govern genomic studies and biobanking activities represents one of the most pressing research-related ethical challenges in a large number of African countries, including Egypt "

o Page 19: " Similarly, many members of RECs in Africa reported strong concerns about the practice of exportation and storage of tissue samples at international institutions, citing fears of the possible exploitation of both African researchers and populations "

o Page 24 :" Lack of clarity about the differences between broad consent and blanket consent was a source of worry among RECs members in African countries"

• Formatting problems: References: some journal names have not been abbreviated properly.

o All references were revised for proper abbreviation.

• Suppl. Tables 2 and 3 run over the page on the right side. Please use landscape page format for wide tables. 

o Done

Reviewer 1

• In the abstract, biospecimens were used throughout whereas samples are being used. I suggest the authors to stick to one word usage and be consistent throughout.

o Biospecimens were used throughout the abstract.

• PAGE 5: What about Intellectual property that comes from the usage of samples for projects.

o Intellectual property has been added in page 4 as follows “These include, among others, questions about privacy and confidentiality, protection of human subjects, benefit sharing, protection of innovation and intellectual property rights", and The following reference was added “Pathmasiri, S., Deschênes, M., Joly, Y. et al. Intellectual property rights in publicly funded biobanks: much ado about nothing?. Nat Biotechnol 29, 319–323 (2011). https://doi.org/10.1038/nbt.1834”

• Page 6:" How many of these centers were targeted?

o Actually what is meant by (center) here is the "central part/region of the country". Sampling was done according to the governorate where the physicians worked. It did not take into consideration the facility/centre where they worked in. The related part of the methods was modified and will now respond to the remark more clearly. 

• Page 6: " The calculated sample size was 218 participants."What would have happened if the calculated sample size weren't reached? Was there a backup strategy in place?

o If the calculated sample was not reached, the strategy was to extend the time limit and increase the communications to recruit more participants.

o Perhaps if these strategies failed, one more governorate from the same three geographic regions could be included.

• Page 15: What about data protection and sharing policies within Egypt and the impact on biobanking activities?

o As a response to this point, the following paragraph has been added in page 20 “At the legal level, a recent data protection law formulated to govern the processing and handling of personal data was proposed by the Egyptian government and approved by Parliament. The Law applies to "personal data" as well as data that can be combined with other data to identify an individual. The law is meant to promote the security of personal data which is being processed and stored online. It also sets a legal framework for the regulation of transmission of data to other countries”

• Page 16: This statement seems to focus more on the aspects of internal governance than external governance. Does this alludes to the fact of lack of national policies related to biobanking.

o Yes, and as we stated in the introduction in page 5 “Although the number of biobanks and their research activities are growing in Egypt, no specific guidelines or regulations for biobanks have been formulated. In Egypt, the only ‘guidelines’ to biobank related research may lie in a number of chapters within a few medical documents”

• Page 16 : Trassport. Spelling mistake. 

o Corrected

• Page 16 : Prisedent. Spelling mistake. 

o Corrected

• Page 17: This is a topic of discussion that always comes up with biobanking discussions and is always swiped under the table and not unpacks. Maybe see how other countries have dealt with these aspects. In SA, our academy of sciences says the following: "5.5 Intellectual Property Rights and Commercialization. The nature and scope of IP rights and the potential for commercialization depend on the outcome of the investigation into whether genomic resources are to be regarded as a common good and thus are outside the domain of private property. If it is assumed that genomic data represent discovery and not invention, the aspect of how IP will accrue from genetic and genomic data is simple. Data need to be ‘productised’ into a service or a product before they can attract IP rights. An example of such a service would be the mining of the data to reveal a specific attribute in the data that was not obvious before. A product would be to use the data to design, e.g. a drug that targets a specific sequencesig nature or a diagnostic test that is designed to detect a specific signature in the sequence. However, a product could also be the compilation of the genetic data in a unique manner. The benefits of commercializing the above service or product will accrue in the bio economy and these benefits are easily distributed if a pre-arrange dmodel exists for its distribution. When genomic and genetic data are used for the purpose of generating benefit, the benefit should be distributed in an appropriate manner, and not just accrue to the public-private entity that innovated with the data. Benefit should reach the group or community whose genetic material was used to generate data" Taken from section 5.5 IP and commercialisation_2018 ASSAF_Human Genetics and Genomics in SA, ethical, legal and social implications consensus study. DOI http://dx.doi.org/10.17159/assaf.2018/0033Tathe data." Taken from 2018_ASSAF_

o Thank you for providing this reference. We used it this part in the text in page 19 as follows “Benefit sharing can provide solution to this dilemma. If genetic data are used to generate healthcare-related benefits, fair distribution of these benefits should be carried out, and these benefits should reach the community which had supplied the genetic material used to generate data resulting in these benefits”

• Page 17: Is there a data sharing and protection law in place in Egypt. If so, this information isn't coming out in the paper. 

o There is a data protection and sharing law as stated above.

• Page 17: It also depends how the data is collected and stored within biobanks. One can have access to clinical information, however only minimal personal information can be shared or it is de-identified for further sharing purposes.

o To clarify what Egyptian biobanks do in this aspect, we added the following statements in page 20 “Protecting the privacy of biobank participants is one of the jobs of biobanks and was stipulated for in the standard operating procedures (SOPs) of Egyptian biobanks. Accordingly, Egyptian biobanks are held accountable for limiting disclosure and safeguarding the integrity of the information stored in them through the use of coding and anonymisation."

• Page 17: Isn't there guidelines from health professional body or government related to different scenario for when confidentiality might be breached. Otherwise this could become a legal nightmare. Intentional and unintentional publication of personal information should also be taken into account. As health professionals we are obliged to adhere to patient confidentiality with sharing only in the capacity of work. 

o We added the following paragraph about this point in page 21 “Article no 30 of the list of professional ethics of the Egyptian Medical Syndicate states that “A doctor is not permitted to divulge the secrets of his patient which he/she had become acquainted with by virtue of his/her profession, unless such divulgence was decreed by a court order or in the event of possible serious and certain harm to others, or in other cases determined by law. However, this applies mainly to medical practice and not research. Thus, conditions where breach of confidentiality during clinical research would be justifiable must be clearly detailed by the syndicate or similar body."

• Page 18: It depends also were the samples collected as part of diagnostic routine and than consented that leftover goes to biobank or was samples specifically donated from researchers. From a legal perspective when a specimen is donated it legally doesn't belong to the participant anymore according to lawyers. But yes Biobanks acts as custodians.

o Thank you for your valuable comment. 

• Page 19: Interestingly throughout the paper nowhere is there reference to Egypt national or institutional ethics committee except regarding approval of the study. One would think whether there is a national or institutional ethics committee there would be guidance set by them with regards to storage of samples as well as the sharing and protection of data. If there is could some regulatory compliance and guidelines refer to them as well as national regulations and acts?

o We added the following paragraph to the introductions in page 5 " Most research ethics committees (RECs) in Egypt use international research ethics guidelines, such as the Declaration of Helsinki and guidelines of the Council for International Organisations of Medical Sciences (CIOMS) and/or the Islamic Organisation for Medical Sciences (IOMS) to review research protocols" and to recommendations in page 26 , which includes data about local ethical committees and network in Egypt “Based on the views of stakeholders and taking into consideration specific factors related to the Egyptian cultural norms and legal framework, we believe that biobanking ethical guidelines would be best developed through the collaboration of different RECs. Such collaboration could be facilitated by the Egyptian Network of Research Ethics Committees, which was founded in 2008 in order to help harmonize the work of different Egyptian RECs. The network includes 23 ethics committees, which can work together to develop an ethical framework for biobanking activities in Egypt."

• Page 19 : So is Broad consent allowed in Egypt? As well as Blanket consent?

o No, only board consent is allowed in biobanks in Egypt. We added this statement in page 24 “Secondly, board consent is the only model of consent that has been adopted and approved by different biobanks in Egypt.”

• Page 20: It all comes down to the consent form whether there is provision made for this. As long as researcher give participants the option regarding return of results. Alternatively a statement by researchers to indicate that return of results will only be in the form of publication and not individually should also be adequate and covered in the consent form.

o Thank you. We added the following statement in page 25 “We believe that, for the time being, return of results should only be offered in the form of publications and not on individual bases. This should be clearly stated in consent forms.”

Reviewer 2:

• Methods section: Authors should add more information on the coverage of the survey. Was a nationally representative database of physicians used, and so the survey was nationally representative? 

o It was mentioned in the methods section that the survey used the convenient sampling technique to recruit the participants. It used the minimum sample size based on the required data by Epi-Info software. The authors selected three governorates which represent three geographic regions in the country (Alexandria representing north, Cairo representing centre and Assiut representing south). There are administratively 27 governorates in Egypt, and more geographic regions could be identified. However our division into north, south and center could be considered a rough classification which is expected to cover governorates with large populations and consequently larger proportions of physicians. The survey was not meant to be nationally representative due to the sampling techniques used. In the modified version of the manuscript we added to the title "limitations" this information and that the data are not nationally representative. 

• The phrase “and communicated with the group...” can be revised for better clarity. What was communicated to the target group? 

o We used the word "distributed" instead of "communicated".

Introduction

General note:

• The authors have written this section in the simplest form, which generally is a good, capturing highlight about biobanking. However, I find a little bit of problem with the synchrony of the paragraphs leading to the justification of why this paper was written. Especially, the link between why stakeholder engagement is needed and thus, this study on stakeholder perception is not fluid enough to support a convincing justification. 

• I seem to think that paragraphs 3 (“One of the biggest….”) and 4 (“Stakeholders can…”) can be modified to make the argument on why stakeholder engagement is needed to support a governance and regulation regime for biobanks in Egypt, which would be expected to further support ethical biobanking research in the country. This new paragraph can then come after the submission about lack of regulations and linked with the sentence that reads: “The development of local ethics guidelines…”. 

o The order of paragraphs in the introduction as been modified. So, the flow will be as follows 

The role of biobanks in genomic research.

The ELSI challenges facing biobanks worldwide, including Egypt.

Biobanking situation in Egypt and lack of specific regulations about it, including ethical regulations

Stakeholder engagement/sustainability as one of the ELSI challenges.

Problem statement and the goal of this work. 

• Paragraph 2, first sentence –“Both genomic research population biobanks are growing….”.There is literature on types of biobanks, which the authors should cite. I have not come across a scholarly reference that has listed “genomic research” as a type of biobank. I am aware of the following types: population-based, disease-specific, tissue, DNA/RNA (or more broadly, “genetic material” and virtual biobanks. All these can support genomics research. A citation to support the biobank types listed by the authors should help. Adjustments may be necessary for the absence of relevant citation to conform with what is more widely documented. Alternatively, the authors may wish to modify the opening sentence of this paragraph. Something like: “Genomic research supported by population biobanks is growing worldwide”. Just a suggestion of how the submission could be better clarified.

o This paragraph in page 4 has been modified as follows “Genomic research supported by population biobanks is growing worldwide, guided by the hope that this research will lead to better disease prevention and treatment through novel drug targeting, personalized treatment, as well as prediction of disease risk. In addition to population-based biobanks, human biobanks can be disease-based (collecting samples from patients with a specific disease or a group of related diseases), genetic (DNA/RNA), project-driven, or tissue versus multiple specimen types. They can be non-profit, commercial, or even virtual (a virtual biobank is an electronic database of biospecimens and data that exists independent of the location of the stored samples) "and a new reference was added “De Souza YG, Greenspan JS. Biobanking past, present and future: responsibilities and benefits. AIDS. 2013;27(3):303-312. doi:10.1097/QAD.0b013e32835c1244

• The first sentence on the page. Consider changing the words “communication” and "Engagement” to their present participle forms for tense agreement with the lead word, "identifying”.

o This has been modified as per your suggestion. 

• Last paragraph before the section on “Subjects and methods”– The sentence that reads: “In our previous work….”, leaves the reader wondering which issues were studied. I would suggest the authors should review this sentence to indicate that in previous work, they reported on knowledge and attitude of patients towards biobanking in Egypt. That building on the report from patients, in this paper, they are reporting on what physicians know and their attitudes towards biobanking.

o This was modified as per your suggestion.

Study design, Participants, Sampling and Setting

• The third sentence that reads “the physicians were recruited…”, can be more clearly written. I would like to see recruitment and questionnaire response as two separate activities. My understanding is that participants were invited to participate in the study via e-mails and through social media. The questionnaire was designed on Google Forms and those that agreed to participate were asked to use the link to the questionnaire provided initially to respond to the survey. Perhaps this much explanation can be moved to the section on “Study too and data collection”, and simply delete the sentence “the physicians were recruited” from this section.

o Modifications in the referred to part of the methods were done as per the advice.

Study tool and data collection

• The second sentence, “study started with an introduction explaining what biobanks are…” is lacking essential details to understand what was done. By saying “the survey started…”, are the authors saying that this bit was contained in the questionnaire, or was it preliminary sort of recruitment information provided? This clarification needs to be made and necessary adjustments to the way it is reported here also made. If it was the former, it would make more meaning to simply highlight the different sections of the questionnaire. If it was the latter, the authors should be clear about that and also indicate how this information was provided to persons recruited via email vs those via social media. Here’s what I am guessing happened. A text containing background information on biobanks and links to other biobank information resources and another to the survey questionnaire was sent either to emails of identified potential participants or a social media handle dedicated for this study. Upon reading the informational text, individuals willing to participate were invited to open the questionnaire link to respond to the survey. The survey questionnaire had mainly three sections – a) demographic information; b) basic knowledge about biobanking before the survey; and c) perceptions and attitudes on biobanking. However, these are my thoughts which I thought I should share so the authors could see why I feel the section, if meaning to convey what my thoughts are, needs to be modified to reflect same. Clarity and synchrony in the conveyance of the information intended are key. Overall, I would like to invite the authors to further review this section for improved clarity and brevity. 

o The introduction was contained in the questionnaire. The paragraph has been modified in page 7 as follows to clarify this point “An introduction preceded the first section of the questionnaire on Google Forms. The introduction included a brief explanation of what biobanks were and a summary of their role in research. This was supplemented by a number of links to videos and sources of further reading about biobanks (supplementary files). The survey questionnaire was comprised of three main sections; a) sociodemographic information; b) basic prior knowledge of biobanking; and c) perceptions and attitudes towards biobanking. Participants were asked to record their level of agreement with survey items (statements) by using a 5-point Likert scale (strongly agree=5, agree=4, unsure=3, disagree=2 and strongly disagree=1).

Ethical considerations

• I do think that the statement provided, which was intended to facilitate an opt-out consent model was inadequate in itself to be accepted and referred to as ‘informed consent’. I would like to understand if the statement referred to by the authors (“By completing this questionnaire…”), was the only information provided to the participants to obtain their consent. Authors should consider writing out summarily how consent was obtained if more than this statement was used in the process.

o The part on (Ethical considerations) was modified according to the above remarks.

• Authors should be more specific highlights on mechanisms adopted to keep data confidential.

o The part on (Ethical considerations) was modified according to the above remarks.

Results

• Page 8, last sentence. It is not clear why the authors chose to classify the participant responses following the last sentence (“Moreover, the following statements…”) as indicating uncertainty among the physicians. Rather I would say the statements show issues for which the physicians were less in favor of regarding biobanking.

o We agree with the reviewer that regarding these four statements the respondents were less in favor with biobanking issues as all the responses were recorded by using a 5-point Likert scale: 5=strongly agree, 4= agree, 3=unsure, 2=disagree and 1=strongly disagree. However, we meant to highlight and specify that around quarter the respondents reported their response as "unsure" which is considered more neutral on the scale compared to "disagree/ strongly disagree). The method of recording the responses to the survey items was added to the subjects and methods section. Subjects and methods/ Study tool and data collection.

• Page 10, paragraph 1. While indeed the sentences and structure in which the results were presented here could be understood, with much attention, I think there is room for improved clarity. I also do not think it was appropriate to group disagreement and uncertainty in the responses. I would suggest that these should be separated since already ‘disagreement’ covered “strongly disagree” and “disagree. If this is done, you would see that except for items 3 and 7, most participants agreed with the statements proffered.

o The authors highlighted the statement which attained the highest level of disagreement (41.3%) and omitted the comment regarding the uncertainty to avoid any probable confusion to the readers. The corresponding paragraph in page 11 was modified.

Page 17

• On the level of agreement on the governance issues: I would suggest we also add the

Statements that responds least agreed with –“Biobanks own the stored samples. (35.4%)”.

o Done and thanks for highlighting these points which will indeed improve the clarity of the results. A sentence was added to the corresponding paragraph in page 12 which addresses the governance issues. 

• On the level of agreement of the respondents to survey items regarding consent, authors should also consider at least reporting the level of agreement on broad consent (47.9%) and expectations to receive some of form of compensation (48.5%). 

o Done and thanks for highlighting these points which will indeed improve the clarity of the results. A sentence was added to the corresponding paragraph in page 13 which addresses the consent and participant's rights in the results section was modified.

Page 18

• On the last sentence on associations between demographic and biobanking KAP variables, authors should consider changing “…and the type of research of current or future research of the participant” to “…and the current or future research interests of the participant”.

o The change suggested by the reviewer was applied in the last paragraph in the results section. 

Page 19

• Knowledge about biobanking – authors should provide the actual proportion of participants that have prior knowledge about biobanks. The first sentence should also be corrected to indicate specifically, the variable that the authors are talking about here – is it knowledge about biobanks in general or that of the existence of biobanks?

o This paragraph has been in page 15 modified as follows " Overall, responses showed that participants had an acceptable degree of knowledge of the meaning and function of biobanks, with 65.5% of participants indicating that they had heard of the term biobanks before " We also indicated the actual proportion of those who know that there are biobanks in Egypt " Despite recognizing the ‘term ‘biobanking,’ less than half of our participants (45.7%) indicated knowledge of the existence of several biobanks in Egypt."

• With a careful review, this section can be markedly improved brevity and clarity-wise. For example, the second sentence here can be modified to read: “In a similar survey among patients, we reported that more than 80% of participants had never heard about the term, “biobanking”, before the survey. 

o This has been modified as per your suggestion.

• The use of the word “that’s” in the tenth sentence is rather informal and should be corrected, please. Authors should also be diligent in making these sorts of corrections in relevant parts of the manuscript. 

o The word has been removed and the whole manuscript was sent for language proofreading. 

Page 20

• General attitude towards biobanking – Authors could modify the sixth sentence to start with “For example, a study similar that being reported here showed a negative correlation…..”.

o This has been modified as follows " Another comparable study to the current work reported a negative correlation between willingness to donate samples to biobanks and higher levels of education"

Page 21

• “Perceptions and attitude…. and sharing of samples”–

Provide relevant reference(s) for the sentence on sharing samples with commercial entities - “Sharing of samples across borders and with commercial….”

o The paragraph has been modified in page 19 as follows “Although commercialization represents one means of achieving financial sustainability, it raises ethical questions regarding fair sharing of benefits and represents a concern for biobank sample donors. Concerns about biobank commercialization are linked to fears of losing control of public biobank resources to for-profit companies and organizations, which may stand in contradiction to the altruistic purposes for which participants volunteer to donate “

And the follwing two refernces were added 

• Nicol D, Critchley C, McWhirter R, Whitton T. Understanding public reactions to commercialization of biobanks and use of biobank resources. SocSci Med. 2016 Aug;162:79-87. doi: 10.1016/j.socscimed.2016.06.028. Epub 2016 Jun 16. PMID: 27343817.)

• Caulfield T, Burningham S, Joly Y, et al. A review of the key issues associated with the commercialization of biobanks. J Law Biosci. 2014;1(1):94-110. Published 2014 Feb 25. doi:10.1093/jlb/lst004)

Page 27

Limitations

• I think the limitations reported are valid, but authors should provide an additional a phrase or a sentence to indicate why the two issues mentioned were limitations. In addition, I would like to encourage the authors to think of other important limitations that they may have missed. For example, was the online-based survey the best approach to optimally reach this sample population? Are there already biases from the approach for this study towards professionals that are social media savvy, and how might that affect representativeness?

o The limitations of the study were modified in page 26 as follows” There have been some limitations to this study. Firstly, we collected data from physicians from three governorates using a convenient sampling method for reasons of time and resource limitations. Although these governorates represent the main geographical regions of Egypt, and medical curricula in different medical schools are comparable, differences in research environments at these medical schools could have affected responses. Secondly, no attempt was conducted to ascertain the amount of bioethics education participants had received before recruiting them into the study. Bioethics education could have affected the responses of some participants, especially with regards to the different types of informed consent. Finally, conducting the survey online may have caused a degree of bias in favor of younger physicians, who are more likely to be comfortable using the internet and social media. Due to these limitations, this study would be regarded as an exploratory study rather that a nationally representative survey."

---

## [Decision Letter · Decision Letter 1]

26 Feb 2021

Knowledge, Perceptions and Attitude of Egyptian Physicians towards Biobanking Issues

PONE-D-20-34081R1

Dear Dr. Abdelhafiz,

We’re pleased to inform you that your manuscript has been judged scientifically suitable for publication and will be formally accepted for publication once it meets all outstanding technical requirements.

Congratulations!

Kind regards,

Helena Kuivaniemi, MD, PhD

Academic Editor

PLOS ONE

Additional Editor Comments (optional):

Reviewers' comments:

Reviewer's Responses to Questions

**Comments to the Author**

1. If the authors have adequately addressed your comments raised in a previous round of review and you feel that this manuscript is now acceptable for publication, you may indicate that here to bypass the “Comments to the Author” section, enter your conflict of interest statement in the “Confidential to Editor” section, and submit your "Accept" recommendation.

Reviewer #2: All comments have been addressed

2. Is the manuscript technically sound, and do the data support the conclusions?

Reviewer #2: (No Response)

3. Has the statistical analysis been performed appropriately and rigorously? 

Reviewer #2: (No Response)

4. Have the authors made all data underlying the findings in their manuscript fully available?

Reviewer #2: (No Response)

5. Is the manuscript presented in an intelligible fashion and written in standard English?

Reviewer #2: (No Response)

6. Review Comments to the Author

Reviewer #2: (No Response)

7. PLOS authors have the option to publish the peer review history of their article (what does this mean?). If published, this will include your full peer review and any attached files.

Reviewer #2: **Yes: **Aminu Yakubu

---

## [Editor Report · Acceptance letter]

9 Mar 2021

PONE-D-20-34081R1 

Knowledge, Perceptions and Attitude of Egyptian Physicians towards Biobanking Issues 

Dear Dr. Abdelhafiz:

I'm pleased to inform you that your manuscript has been deemed suitable for publication in PLOS ONE. Congratulations! Your manuscript is now with our production department. 

Kind regards, 

on behalf of

Professor Helena Kuivaniemi 

Academic Editor

PLOS ONE